# Transfer Learning Applied to Characteristic Prediction of Injection Molded Products

**DOI:** 10.3390/polym13223874

**Published:** 2021-11-09

**Authors:** Yan-Mao Huang, Wen-Ren Jong, Shia-Chung Chen

**Affiliations:** Department of Mechanical Engineering, Chung Yuan Cristian University, Taoyuan City 320314, Taiwan; H05850207@hotmail.com (Y.-M.H.); shiachun@cycu.edu.tw (S.-C.C.)

**Keywords:** injection molding, CAE, machine learning, transfer learning

## Abstract

This study addresses some issues regarding the problems of applying CAE to the injection molding production process where quite complex factors inhibit its effective utilization. In this study, an artificial neural network, namely a backpropagation neural network (BPNN), is utilized to render results predictions for the injection molding process. By inputting the plastic temperature, mold temperature, injection speed, holding pressure, and holding time in the molding parameters, these five results are more accurately predicted: EOF pressure, maximum cooling time, warpage along the *Z*-axis, shrinkage along the *X*-axis, and shrinkage along the *Y*-axis. This study first uses CAE analysis data as training data and reduces the error value to less than 5% through the Taguchi method and the random shuffle method, which we introduce herein, and then successfully transfers the network, which CAE data analysis has predicted to the actual machine for verification with the use of transfer learning. This study uses a backpropagation neural network (BPNN) to train a dedicated prediction network using different, large amounts of data for training the network, which has proved fast and can predict results accurately using our optimized model.

## 1. Introduction

In recent years, the computer, communications, and consumer electronics industries (hereafter referred to as the 3Cs) have developed vigorously, and their increasing diversity of products has led to the shortening of product life cycles. In order to reduce a costly trial-and-error process and speed up product development to match product life cycle demand, many companies have introduced computer-aided engineering (CAE) technology to their molded products’ production processes in order to enhance the quality of those products, ideally avoiding defects. However, due to the physical characteristics of each injection machine being irregular, even though possible problems are eliminated at the CAE stage, there is still a necessity to rely on the experience of on-site personnel to manually adjust the parameters of said machines for the actual molding process to be carried out successfully. As such, in order to achieve the accuracy required for a product, CAE technology alone cannot currently provide sufficient assistance regarding such processes requiring a precise injection of molding to create products.

When defects do occur in products created via an injection molding process, these are not due to simple linear problems. Therefore, there is no way to use simple judgment rules to predict numerical values. In related research literature, Rosa et al. [1] mention that experimental design, i.e., design achieved using experimental techniques, is widely used to optimize injection molding parameters, thus improving product quality. However, conventional experimental design methods are usually complicated and may often fail to achieve the results expected. When the number of molding parameters increases, these methods require many trials. (It can be said that as the number of parameters to be included increases, the number of trials necessarily increases.) Therefore, the Taguchi orthogonal method is used to select experimental trial data and used to reduce the number of trials required to obtain clear results; Marins et al. [2] propose the use of the Taguchi method and analysis of variance (ANOVA) to evaluate the impact of varied injection molding parameters regarding warpage, shrinkage, and mechanical properties of plastic parts. Marins et al. use two different plastics, one of which crylonitrile butadiene styrene (ABS) is used in this study. Their control factors are mold temperature, holding pressure, holding time, plastic melt temperature, cooling time, mold water flow, and injection speed. The results of their experiments show that when ABS was used for trials, the controlling factors regarding shrinkage, warpage, and bending defects are holding time and holding pressure.

Hifsa et al. [3] use the Taguchi method with grey relational analysis to find the best parameter configuration for injection molding of HDPE/TiO_2_ nanocomposites.

In another related work on machine learning, Luo et al. [4] employ an artificial neural network to resin transfer molding (RTM) using simulation analysis results as training data and flow behavior and filling time as output, combined with a genetic algorithm to optimize the prediction results. Their research results show that this method can effectively assist the engineer to determine the optimum locations of injection gates and vents for the best processing performance, i.e., short filling time and high quality (minimum defects).

However, Kenig et al. [5] mention that the mechanical properties of plastics and molding parameters are highly nonlinear. Therefore, they are difficult to predict. Kenig et al. use the design of experiment (DOE) method in combination with artificial neural networks to accurately predict the mechanical properties of the product. This method can be used to predict other molding results effectively. 

Moreover, Denni [6] proposes the use of the Taguchi method, a backpropagation neural network (BPNN), and the hybrid particle swarm and genetic algorithms to find optimal parameter settings. The results of Denni’s experiments show that this optimized system not only improves the quality of plastic parts but also effectively reduces process variation. Denni also mentions that a backpropagation algorithm will cause difficulty in reaching optimal solutions or overfitting due to poor initial link values or excessive training numbers. Therefore, genetic algorithms are added to alleviate this shortcoming and increase the accuracy of predictions. 

Furthermore, Kwak et al. [7] propose a kind of artificial neural network architecture to solve the multivariable problem that affects the optical mold during the injection molding process using the two control factors of suppressing porosity and reducing the minimum thickness combined with an artificial neural network to make predictions, which proved that this technology can effectively improve the product quality of optical molds.

In addition, Castro et al. [8] assert that injection molding is the most important process for mass production of plastic products; however, the difficulty in optimizing the injection molding process lies in performance measurement. Therefore, CAE, statistical methods, artificial neural networks (ANN), and data envelopment analysis (DEA) are several methods to solve the problems encountered in injection molding. 

In their work, Shen et al. [9] mention that injection-molded products are sensitive to the conditions of the injection process, so they use backpropagation to process the nonlinear relationship between the process parameters and product quality and combine genetic algorithms to optimize the process parameters to reduce product shrinkage. Their results show that the combination of backpropagation and genetic algorithms can be an effective tool for optimizing process parameters.

Mirigul [10] uses the Taguchi method to carry out experimental design and ANOVA to perform analysis, finding that the most important factors that affect polypropylene (PP) and polystyrene (PS) shrinkage are holding pressure and melt temperature, and through these trials, with melt temperatures, holding pressures, holding times used as input, and the shrinkage rate used as output, an artificial neural network training is conducted. Mirigul’s research results show that PP shrinkage prediction error is 8.6%, and for PS, it is 0.48%, which proves that this method is an effective tool for predicting shrinkage.

Additionally, Yin et al. [11] propose a 5-input nodes, 1-output node backpropagation for neural network learning. The 5 input signals are mold temperature, melt temperature, filling pressure, filling time, and holding time. The hidden layer contains two neural layers, each with 20 neurons, and the output signal is warpage deformation. This method successfully uses finite element analysis of data for training and the prediction error of the control systems is within 2%.

Alberto and Ramón [12] mention that the process parameters are one of the final yet important steps during production; they are used to improve product quality in final production, but the adjusted parameters might affect the quality of a product due to the instability of material and machine. Instability affects product quality. From their work, we can see that using machine learning can accurately improve stability and quickly improve product quality.

Deng and Yu [13] propose general deep learning methodologies and provide an overview of their application to various signal and information processing tasks. Per their work, there are three criteria for selecting an application area: the author’s professional knowledge, the application areas that have been successfully transformed by the use of deep learning technology, and the application areas that are potentially affected by deep learning. Their monograph also introduces the principle of pretraining in unsupervised learning.

In the work of Jong et al. [14], the hyperparameters of a backpropagation neural network (BPNN) are optimized with the smaller, the better feature (STB) from the Taguchi method, and they propose a 5-input, 3-output artificial neural network architecture, where the 5 inputs are injection speed, holding time, holding pressure, mold temperature, melt stability, and the 3 outputs are end of filling (EOF) pressure, maximum cooling time, and warpage along the *Z*-axis. There are two hidden layers: the first layer has 7 neurons, and there are 3 neurons in the second hidden layer. The network is trained for the second time with new training data for addressing Z-axis warping, and in terms of the warping deformation value, the error is reduced from 7.26% to 3.69%.

Sinno et al. [15] propose that between classification tasks in similar fields, transfer learning, if performed successfully, would greatly improve the performance of learning by avoiding expensive data-labeling efforts.

In the research literature related to transfer learning, Dan et al. [16] use an artificial neural network that recognizes various symbols, and it is retrained with the data of capital letters. The results show that using the original network for transfer learning can accelerate the training efficiency of the neural network. Furthermore, pretrained neural networks consistently outperform randomly initialized neural networks on new tasks with few labeled data. This result is also applied to Chinese character recognition for accelerated neural networks.

Huang et al. [17] propose a sharing cross-language hidden layer concept for the learning of various languages, and this hidden layer of cross-language learning features transformations during transfer learning in which the error can be reduced to 28% at best.

Jiahuan et al. [18] introduce model-based transfer learning and data augmentation, the knowledge from other vision tasks is transferred to industrial defect inspection tasks, resulting in high accuracy with limited training samples. Experimental results on an injection molding product showed that the detection accuracy was improved to about 99% when only 200 images per category were available. In comparison, conventional CNN models and the support vector machine method could achieve an average accuracy of only about 88.70% and 86.90%, respectively.

Yannik, L. et al. [19] used induced network-based transfer learning to reduce the necessary amount of injection molding process data for the training of an artificial neural network in order to conduct a data-driven machine parameter optimization for injection molding processes. From the research results, it is found that the source model of the injection molding process of the part similar to the part of the target process achieves the best result. Transfer learning technology has the potential to significantly improve the relevance of AI methods in process optimization in the plastics processing industry.

In their work, Shin et al. [20] mention that because ImageNet collects a large number of image tags, the neural network acquires enough training data to make accuracy ever-improving. Their research results show that ImageNet network transfers to thoracic–abdominal lymph node (LN) detection and interstitial lung disease (ILD) classification problems with good accuracy, proving that neural networks can be extended to other medical applications through transfer learning.

Hasan et al. [21] mention that when the amount of training data is not sufficient when using training with random initialization, it is difficult to find an optimal solution. This research first pretrains the network with simulated and analyzed data and then performs transfer learning with actual experimental training. Their research results show that this pretrained method can make a network converge quickly and can obtain results similar to those of the control group with less training data. In related work, Hasan et al. [22] mention that machine learning has great potential in the injection molding process, and their research involves using different masses of LEGO bricks as training data and then transferring what is learned to LEGO bricks of other sizes for quality prediction. The network used in their study contains four hidden layers of neurons, the numbers of which are 45, 45, 20, and 20, respectively. Rectified linear unit (ReLU) is used as the activation function, the learning rate is 0.01, and the first two neural layers are frozen during the transfer learning process to ensure that important knowledge will not be lost during the transfer process.

From the aforementioned works, it can be found that in the application of the injection molding process, most research is used to predict product weights, and less focus is given to injection pressure, product shrinkage, warpage, or other issues. Additionally, beyond the issue of weight, another criterion for judging whether the injection molding process for any given product meets the standard is the shrinkage rate. At present, 3C products are becoming more and more complex, so the assembly accuracy of the product is demanding, making the parameters related to the shrinkage rate complicated. The injection speed of the machine, mold temperature, holding pressure, and holding time are directly related. Therefore, the on-site staff cannot quickly determine how to adjust the production process to meet the tolerance specified during the design. The transfer learning method proposed herein trains an artificial neural network with fewer actual parameters, which can be used in the future to assist mold testing staff to quickly adjust machine parameters.

## 2. Relevant Technical Research

In recent years, with the ever-increasing speed of computer operations, artificial neural networks have once again been widely discussed. With the development of TensorFlow as an open-source resource by the Google Development Team, the application of artificial neural networks has begun to develop rapidly. Artificial neural networks can be “trained” with a significant amount of data constructing a network with rapid prediction and judgment capabilities. However, not all trials can easily have a large amount of training data collected. Therefore, the concept of transfer learning is also proposed. Neural networks trained through similar trials can reduce the amount of training data and time necessary to construct new networks.

Regarding programming development, this research uses the programming language Python to interface with API via TensorFlow 2.0 (a Python-friendly open source library) to develop artificial neural networks. After the release of TensorFlow version 2.0, the high-level API highly supports Keras to reduce the difficulty of getting started and enables interdisciplinary researchers to construct neural networks and adjust network hyperparameters more conveniently.

### 2.1. Artificial Neural Networks (ANN)

Artificial neural networks are currently widely used in the field of machine learning and artificial intelligence. An artificial neural network consists of an input layer, a hidden layer, and an output layer. Each layer is composed of several neurons. This study utilizes a backpropagation neural network as a training method, and it adjusts the values of weight and bias between each neuron based on the results of each training trial.

#### 2.1.1. Backpropagation Neural Network (BPNN)

A backpropagation neural network (BPNN) is a kind of supervised learning that uses the gradient descent method to correct the erroneous difference between the expected value and the output value. Its characteristic is that each neuron has a differentiable activation function, and the network is highly connected.

The curve of injection molding product has multiple and deformable characteristics. It is hard to use a simple principle to predict. Therefore, this research is based on BPNN to predict these irregular data.

The BPNN network structure can be divided into three layers. As shown in Figure 1, the first layer is the input layer. The input layer will receive data from the outside world. Usually, these data will be preprocessed first. After receiving the data, the input layer will transmit the data to the next layer. The number of hidden layers and the number of neurons in each layer are designed according to different modeling problems, and because each neuron is highly connected to each other, the architecture of the hidden layer will directly affect the calculation speed of each training run; the third layer is the output layer, the predicted values calculated by the hidden layer are exported to the outside world from this layer, and according to different activation functions, these will have different functions. Common ones are numerical prediction, classification, and probability, and so on.

#### 2.1.2. Training and Learning Process

A backpropagation neural network is trained through the use of multiple sets of data and continuously adjusts the weight value of each link until the error of the network output meets the expected range. The training process is characterized by forward flow of information in the prediction mode and backward flow of error corrections in the learning process. Usually, the initialized weight value and offset value are randomly generated, so the data need to be passed in the forward direction. The input data passed to the output layer after the weight value, offset value, addition function, and activation function of the link are calculated and predicted values can be received; the reverse signal transmission is the reverse transmission stage of the network training. The error between the network’s predicted value and the target value is first calculated, and then the learning effect is controlled by the gradient descent method using the learning rate. The learning process is shown in Figure 2.

### 2.2. Transfer Learning (TL)

The concept of transfer learning originated from the study of human behavior by scientists. Human beings can transfer past experience to different fields and thereby accelerate their learning in those fields new to them. In the field of machine learning, the artificial neural network algorithm was developed by simulating the operation of human nerves, and it also has the characteristics of transfer learning. When we do not have enough labeled data to train the model, we can use a similar or the same type of data to “pretrain” a model; with and on the basis of this properly trained model, a usable model can be trained by using less labeled data. Transfer learning can be divided into two types—first, the transition from virtual to actual reality, and second, among similar areas (as is explained shortly hereafter). If the acquisition of training data is overly risky, the amount of data required is too large, or it takes a lot of time and money, using a computer simulation can obtain sufficient training data in a fast, safe, and cost-effective manner. For example, if automatic driving technology were directly applied to a real road test before it is mature, it would be extremely dangerous. Therefore, the use of computer-simulated driving for a road network as pretraining and then transferring that to actual road training will greatly reduce both risks and data requirements. Another transfer method is to transfer to a similar field. For example, the module used for truck identification can be quickly applied to sedan identification, which not only reduces the training data required but also speeds up the network training. To achieve an effective transfer of learning, the data selection must be as similar as possible, and, if possible, the data should be adjusted to match the data type of the original network. Therefore, the data must be guaranteed to be normalized without too much deviation; otherwise, the content received by the network will be different from the setting.

### 2.3. Computer-Aided Engineering (CAE)

Computer-aided engineering (CAE) is the use of finite element software to establish an analysis model and perform rapid calculations on a computer for simulation. It is mainly used in various engineering fields. Using the injection molding industry as an example, engineers can perform simulation analysis via computer at an early stage of product design and then evaluate the feasibility of that product design, as shown in Figure 3. This technology can greatly shorten the development time, analyze and correct designs at the initial stage, and reduce manpower and material costs in the subsequent testing process so as to accelerate the time of product development, improve the product yield and quality, and then increase the yield.

### 2.4. The “Random Shuffle” Method

In machine learning, the most important thing is the correctness and quantity of training data. Sufficient training data is one of the primary factors to ensure accuracy, but how to use such hard-earned data is another issue of importance. Although machine learning is fast and accurate, it never makes mistakes, so no matter how many calculations are processed, the same result will be obtained. When humans are learning, because of factors such as distractions and different preferences, the same information is evaluated differently, which increases the likelihood of different resultant possibilities. Therefore, when processing training data, that data can be randomly grouped, as shown in Figure 4, by splitting it into multiple sets for learning purposes, the possibility of optimization can thus be enhanced. This method, which we coin here as “random shuffle,” is similar to the concept of training on batch commonly used in machine learning. “Training on batch” describes when a training data set is divided into multiple batches in order to avoid the difficulty of convergence due to processing too much training data at a time. The greatest difference between training on batch and random shuffle is that the former uses the data in only one training run, while random shuffle utilizes incomplete data sets for each run, thus increasing the chance of mutation during all training runs. Because each training data set is incomplete, after training, you need to train it again using the complete data set. Random shuffle can be regarded as a kind of pretraining, and then the full data are used to make final adjustments to the pretraining network, such as shown in Figure 5, where the data segmentation of random shuffle needs to be searched for using the trial-and-error method. This research has been tested at 80% of the full data segmentation with a learning rate of 0.1, which shows an improved effect. In terms of epochs, the number depends on the amount of data in the set—the more training data that are in the set, the fewer epochs will be needed.

### 2.5. Data Normalization

Data normalization is used to compare data from different units. A unit of a set of data can be removed and analyzed. Taking the injection molding parameters applied to neural network training as an example, a temperature of 200 degrees for a plastic and the holding time of 5 s as the input layer, due to the high value of the plastic’s temperature, more iterations will be required in the calculations with the gradient descent (algorithm), which will cause too many iterations on the holding pressure intervals with a relatively small value that, in turn, causes overfitting. Therefore, the preprocessing of data is important for machine learning. The two most common methods in the field of machine learning are min–max normalization and *Z*-score standardization.

The main purpose of minimum and maximum normalization is to scale all the data to between [0, 1] and retain the distribution state of the data. The calculation method is shown in Equation (1). This method is mainly used when the difference among data sets is too great, which can accelerate the convergence of the model and avoid potential overfitting; Z-score standardization is used to convert the data and to make it conform with normal distribution, so that the average value is equal to 0 and the standard deviation is equal to 1. The calculation method is shown in (2); it is mainly used to further optimize data when the data themselves have close to a normal distribution. The data used in this study are all evenly distributed, and the numerical difference between each input is large, so the minimum and maximum normalization will be used to perform the preprocessing of the data.
(1)$$X_{nom}=\frac{X-X_{min}}{X_{max}-X_{min}}\in \left[0,1\right]$$
(2)$$Z=\frac{X-\mu }{\sigma }\sim N\left(0,\;1\right)$$

### 2.6. The Taguchi Method

The Taguchi method is a statistical method developed by Dr. Taguchi Genichi in the 1950s. It can improve design quality and computational cost efficiency. Through the use of an orthogonal meter, the interference of noise on the product can be reduced, the number of trials can be reduced, and the quality variation can be reduced to achieve the purpose of robust design.

#### 2.6.1. Quality Characteristics

Appropriate quality characteristics have the following two criteria: one is that real, continuous functions are required, and the other is that a monotonic function is preferred. It may likely be anticipated that “smaller” error between the inferred value and the target output value is better, so it has a quality characteristic of “the smaller, the better” (STB), and for STB, its target value of the *S/N* ratio should be approximating 0. The definition of the *S/N* ratio is shown in Equation (3).
(3)SN=−10log[1n∑i=1nyi2]

#### 2.6.2. Definition and Selection of Experimental Factors

The factors that affect quality characteristics can be divided into three categories, namely control factors, signal factors, and interference factors (also called noise factors), as shown in Figure 6. The quality of the design will affect whether the subsequent results achieve the standard desired and will not change with excessive variation due to the interference of external factors, that is, sensitivity to the interference factor is reduced.

Three strategies can be used for interference factors: random experiment, internal orthogonal table, and interference experiment. If the quality characteristics are still not affected by the extreme compound interference factors, then the combination is the best stable design and can resist changes from other interference factors. To understand the Taguchi orthogonal method as mentioned prior, the main effect between the factors, that is, the degree of influence of the factor on the quality characteristics, and the fact that the average effect of each factor is the experimental combination configured in the Taguchi orthogonal table experiment, must be understood and ascertained, and then the *S/N* ratio of each factor calculated, and finally made a response table.

## 3. Using CAE Data to Study Transfer Learning among Different Models

Because of the fact that actual injection molding data is difficult to obtain, this study first uses CAE software to analyze a circular flat plate as the source of training data and combines our random shuffle and database normalization processes for network training and optimization; then, the trained network will be transferred and used on a similar but square plate model. Finally, the use of the CAE data of the square plate model is used to readjust the transferred weight and bias to make the entire network more suitable for the numerical prediction of the square plate, as shown in Figure 7.

### 3.1. Network Training of the Circular Flat Model

This study uses a circular flat panel model to perform the pretraining of the backpropagation neural network (BPNN). The pretraining is divided into two parts: data collection and network parameter tuning. The pretraining is to provide a set of available initial weight and bias for the square flat panel model as a training purpose.

#### 3.1.1. Training Materials

The training data are produced by using the mold flow analysis software, Moldex3D. Each set of training data contains 5 control factors as input values and 5 analysis results as expected values of output. The input values are injection speed, holding pressure, holding time, plastic temperature, and mold temperature; the output values are gate EOF (end of Filling) pressure, maximum cooling time, the value of warpage along the *Z*-axis, product shrinkage along the *X*-axis, and product shrinkage along the *Y*-axis. The measurement points are shown in Figure 8.

In terms of the process parameters, this study takes the default setting provided by Moldex3D as a reference and then set values according to the number of levels. Among the 5 control factors, the injection speed, melt temperature, mold temperature, holding time, and holding pressure are set at 3 levels, and then using a full factorial trial, a total of 243 sets of training data will be produced. The detailed training data parameters are shown on Table 1. After the artificial neural network training is completed, it needs to be tested with data that has not been learned at all. In this study, 16 sets of parameters corresponding to the L16 orthogonal table of the Taguchi method are used as verification data. The detailed parameters are shown on Table 2.

#### 3.1.2. Hyperparameter Settings

In terms of artificial neural networks, this study uses the optimized network parameters proposed by Jong et al. [14]. The parameters that need to be set are the number of training times, the number of repetitions, the activation function, the learning rate, the optimization method, the initial link value, the number of hidden layers, and the number of neurons; this network architecture is shown in Figure 9. As for other settings, the number of training runs is reduced to 20,000 due to the introduction of random shuffle for pretraining, and the learning rate is 0.1. Both hidden layers use sigmoid as the activation function and use the stochastic gradient descent (SGD) for optimization, and the network hyperparameter settings are shown on Table 3.

As for the data input, this research uses random shuffle to randomly divide 243 process parameters into 1000 sets of training data. Each set of data contains 195 (80%) process parameters. The network alternates a set for training every 500 trials during training to avoid overfitting occurring.

### 3.2. Transfer Learning of the Square Plate Model

In terms of transfer learning, the square plate and the round plate model are the same, as are the network architecture of the two, as well as the optimized weight and bias values for both, and then the training data of the square plate are used to fine-tune the network to achieve the goal of transfer learning.

#### 3.2.1. Training Materials

The training data are the same as those of the round flat model. The five analysis results are used as the expected values of output. The input values are injection speed, holding pressure, holding time, plastic temperature, and mold temperature; the output values are EOF pressure, maximum cooling time, the value of warpage along the *Z*-axis, the amount of product shrinkage along the *X*-axis, and the offset of the product along the *Y*-axis. The measurement point positions are shown in Figure 10.

The purpose of transfer learning is to reduce the required training data and save calculation time. Therefore, in the training data setting for the square tablet, the L27 orthographic table of the Taguchi method is used to produce 27 sets of training data as input, as shown on Table 4. The L16 orthogonal array produces 16 sets of verification parameters to verify the network, as shown on Table 5.

#### 3.2.2. Hyperparameter Tuning

The hyperparameters used in the transfer learning of the square plate are the same as those used for the round plate. The architecture uses two hidden layers. The first layer contains seven neurons, and the second layer contains three neurons. For other settings, the learning rate is 0.1. Both hidden layers use sigmoid as the activation function. A transfer function is used to input weight and bias. The network hyperparameter settings are shown on Table 6.

As for the data input, this research uses random shuffle to randomly divide 27 sets of process parameters into 1000 sets of training data. Each set of data contains 22 (80%) process parameters. The network alternates a set of training for every 500 trials during training to avoid the occurrence of overfitting.

### 3.3. Comparison of Transfer Learning Results among Different Products

This study compares the training process of transfer learning with full data and random shuffle processing and discusses the results of transfer learning based on the two results of the *S* mean error and standard deviation.

#### 3.3.1. Training Results of the Round Plate Model

In this study, the prediction results of 243 groups of circular flat models analyzed via Moldex3D were used as the input value for the backpropagation neural network, and the data processing method of random shuffle was utilized to improve accuracy of prediction. With the same training data, network architecture, and hyperparameter settings, after using the data processing method of random shuffle, the average error of EOF pressure, cooling time, warpage value along the *Z*-axis, shrinkage along the *X*-axis, and shrinkage along the *Y*-axis all delivered different degrees of improvement, of which *Z*-axis warpage shows the most significant mitigation. In addition to reducing the average error, random shuffle also has different degrees of optimization vis-à-vis the standard deviation. The standard deviation regarding warpage has been reduced most obviously. Our detailed comparison results are shown on Table 7.

#### 3.3.2. Training Results for the Square Flat Model

In the case of trials on the square plate, this study only uses 27 sets of training data as the input value for the backpropagation neural network, and the process parameters are the same as those for the round plate. Our detailed training results are shown on Table 8. If 27 sets of data are used for training directly, among the five output values, only EOF pressure and cooling time are having less than 10% of predicted error. As for warpage along the *Z*-axis, shrinkage along the *X*-axis, and shrinkage along the *Y*-axis, their error is high, of which the 59% for the warpage along the *Z*-axis is the highest. After using random shuffle to reprocess the data, the error rate and standard deviation of the five output results are all reduced, but the errors of warpage and shrinkage along the *X* and *Y* axes are still significant.

#### 3.3.3. Transfer Learning Results for the Square Tablet Model

The round flat and square flat are both flat types of models. Therefore, this study uses the trained round flat network architecture as the network architecture for square flat training and integrates the optimized weight and bias into the network as the starting values for square tablet training. Our training results are shown on Table 9. Without preprocessing the data, using the optimized weight and bias from training for the round flat plate, the error value and standard deviation of the EOF pressure, cooling time, shrinkage along the *X*-axis, and shrinkage along the *Y*-axis are all reduced, except for the average of percentage errors for warpage along the *Z*-axis, which has increased from 59.61% to 79.96%; if the data are preprocessed with random shuffle, the error value and standard deviation of the five output items will decrease, among which the warpage has been mitigated most obviously, and the error value has been reduced from 79.96% dropped to 31.05%, and the standard deviation dropped from 70.34% to 17.56%.

## 4. Prediction of Molding Using the Network Trained via CAE Data and Transferred for Actual Injection

In this section, the use of a larger amount of circular flat model CAE data to train a more accurate neural network is described and discussed. This network trained with CAE data is then transferred to the actual injection molding prediction, and the circular flat data obtained from actual trials are used for retraining to verify that there is feasibility of conversion between virtual and actual reality models for the injection molding process.

### 4.1. Transfer Learning between Virtual and Actual Reality

In the transfer learning process between virtual and actual reality, because the same model is used, the two sets or types of data must be as similar as possible. If the actual data and the CAE data are too divergent, prediction accuracy is seriously impacted after the final transfer. Therefore, the data collection during the running of trials is important. In this study, with the gates, the in-mold pressure sensor was used to collect measurements of the pressure change curve. After the product was left for one day, the round plate of the amount of change in the *X*- and *Y*-axis was measured using a coordinate measuring machine; however, warpage and cooling time data are difficult to obtain, so these two results are not considered during the transfer process. The entire process is shown in Figure 11.

### 4.2. Round Flat CAE Data Pretraining

The prediction accuracy before transfer of learning greatly affects the results after transfer. Therefore, herein, the use of more CAE data and the reoptimization by tuning the hyperparameters using the Taguchi method and the performance of pretraining before TL are detailed and discussed. All prediction errors of the training model are maintained at less than 5% of the total.

#### 4.2.1. Training Data

The training data in this section are produced by the mold flow analysis software, Moldex3D. The number of data sets has been increased from 243 sets to 1024 sets. Each set of training data contains 5 control factors as input values and 5 analysis results as expected output values. The input values are injection speed, holding pressure, holding time, plastic temperature, and mold temperature, respectively; and the output values are gate EOF pressure, maximum cooling time, warpage along *Z*-axis, shrinkage along the *X*-axis, and shrinkage along the *Y*-axis. The position of the measuring point is shown in Figure 12.

In terms of process parameters, for this study, the preset value provided by Moldex3D is taken as the reference, and then the value is adjusted according to the number of levels. Among the five control factors, the injection speed, melt temperature, mold temperature, holding pressure time, and holding pressure are all set at level 4, and then using the full factor method, a total of 1024 sets of training data are produced. Our detailed training data parameters are shown on Table 10. After the artificial neural network training is completed, it needs to be tested with data that have not yet been learned. In this study, 16 sets of parameters corresponding to the L16 orthogonal table of the Taguchi method are used as verification data. Our detailed parameters are shown on Table 11.

#### 4.2.2. Hyperparameter Tuning

In order to achieve better prediction accuracy, 1024 sets of data are used in this section and 1024 process parameters are randomly divided into 200 sets of training data with random shuffle. Each set of data contains 819 (80%) processes. A set of training materials is changed every 50 runs during training to avoid overfitting.

This study uses the optimized network architecture proposed by Jong et al. [14] to train 1024 sets of CAE data and found that if the warpage along the *Z*-axis cannot be less than 5%, loss value cannot continue to decrease, even with the number of training runs increased; for this, the change of loss value can be observed from either the training data or verification data. If overfitting is the cause of the inability to reduce accuracy, the loss values of the two (i.e., the loss value of the training and verification set) will cross over. However, in fact, the loss values of the two, though they fluctuate significantly, remain parallel, and there is no crossover phenomenon, as shown in Figure 13. The rapid rises and falls, seen illustrated in Figure 13, represent that weight and bias are continuously being updated, but the loss value cannot be further reduced. Therefore, it can be inferred that in the optimized network architecture proposed by Jong et al. [14]., the number of neurons cannot fully learn the knowledge presented by the 1024 sets of data. Therefore, in this section, discussion of use of the Taguchi method to recorrect the optimized network design and hyperparameters is present, and factor levels are shown in Table 12, while the orthogonal array is shown in Table 13.

#### 4.2.3. Hyperparameter Optimization

In this section, with the use of random shuffle, 200 sets of new data are made with 80% of the full data (819 sets of data) are detailed and discussed. Because each set of data is relatively large, in order to avoid overlearning the content of a single set of data, the epoch is set to 50, and the learning rate is 0.1 for pretraining, hyperparameter optimization, using *Z*-axis warpage as the indicator of optimization as discussed prior in Section 3, and the error percentage between the predicted value and the expected value is used as the optimization target (i.e., STB). Trials results from the Taguchi orthogonal array are shown on Table 14. The most effective result is No. 5, the warpage along *Z*-axis is 5.22%, and our ANOVA analysis is shown on Table 15. The optimization factors are A2, B1, C3, D1, and the corresponding hyperparameters are as follows: training times 20,000, learning rate 0.05, 11 neurons in the first layer, and 7 neurons in the second layer, and the difference between the optimized parameters and No. 5 is only the factor B, which are the learning rates of 0.05 and 0.1, respectively.

It can be found from Figure 14 that the C and D factors representing the neural network structure in the ANOVA analysis, in the indications, are the same as the results given prior in Section 3, which is the first layer is the bigger, the better, and the second layer is the smaller, the better, and the first layer is larger than the second layer; as for the learning rate, it shows a trend that the smaller the value, the better, so this research uses this trend to adjust the optimized hyperparameters to 20,000 training times, learning rate 0.01, 13 neurons in the first layer, and 5 neurons in the second layer. Finally, this parameter with No. 5 and the optimized parameters are compared. The results are shown on Table 16. The optimized parameters have the smallest error value. The combination adjusted with optimized parameters is also better than No. 5. The network architecture is shown in Figure 15, and the hyperparameters are shown on Table 17.

### 4.3. Transfer Learning for Injection Molding

In the transfer learning between virtual and actual reality, the data collection of the actual reality trials is important. The final results will be more accurate if the CAE data are more similar. For this study, the training data and verification data are obtained by actual injection molding, and then the training data are imported to the pretrained model.

#### 4.3.1. Training Data

In terms of the training data for the injection molding process, the input values are injection speed, holding pressure, holding time, plastic temperature, and mold temperature, and the five analysis results are categorized as the expected output values, respectively. As for the output values, because it is difficult to measure the amount of warpage along the *Z*-axis and the maximum cooling time, so for this study, the output values of EOF pressure, shrinkage along the *X*-axis, and shrinkage along the *Y*-axis only are captured. The other two are replaced by 0 during training, so that the network can run smoothly.

The training data and verification data discussed herein are all based on the L16 OA table, each generating 16 sets of data and a total of 32 unique data sets. Due to the gap between the actual machine settings and the simulation software, the holding pressure is set at the upper limit of 140 MPa as the basis to modify the molding parameters. The relevant process parameters are shown on Table 18 and Table 19.

#### 4.3.2. Hyperparameter Settings

The hyperparameters used in transfer learning are the same as those used in CAE data pretraining. In terms of the architecture, two hidden layers are used. The first layer contains 11 neurons, and the second layer contains 7 neurons. For other settings, the learning rate is 0.05. Both hidden layers use sigmoid as the activation function. Weight and bias use the transfer method. The network hyperparameter settings are shown in Table 20.

As for data input, this research uses random shuffle to randomly divide 16 process parameters into 1000 sets of training data. Each set of data contains 13 (80%) process parameters. The network changes one set of training data after every 500 runs during training to avoid overfitting from happening.

### 4.4. Comparison of Simulation and Actual Transfer Learning Results, CAE Data Training Results for the Circular Flat Model

In this study, the prediction results for the 1024 sets of the circular flat model analyzed by Moldex3D were preprocessed with random shuffle data processing as the input value of the backpropagation neural network and combined with the hyperparameters generated using the Taguchi method to optimize the model. After these methods are used, the EOF pressure, cooling time, gate warpage value, shrinkage along the *X*-axis, and shrinkage along the *Y*-axis, the average error is less than 5%, and *Z*-axis warpage is mitigated most obviously, which is a more complicated factor. In addition to reducing the average error, random shuffle also has different degrees of optimization vis-à-vis the standard deviation. In terms of the improvement in the standard deviation, gate warpage is the most obvious. Our detailed comparison results are shown in Table 21.

#### 4.4.1. Training Results of Experimental Data of the Circular Flat Model

In the actual trials, this study only uses 16 sets of data for training and 16 sets of data for verification. The hyperparameters are the same as the CAE data for training. The detailed training results are shown in Table 22. If 16 sets of data are used for training directly, all 3 output values are not ideal. After using random shuffle to reprocess the data, the error rate of the three output results has slightly decreased, and the standard deviations have shown significant reduction.

#### 4.4.2. The Results of the Transfer Learning Trials Data for the Round Plate Model

In terms of data collection, the CAE and actual trials use the same model. In this study, the circular flat network structure trained with CAE data is used as the network structure for the trial data, and the optimized weight and bias are imported as training input. The training results are shown on Table 23.

Without preprocessing the data, this study uses the optimized weight and bias of the CAE data for training, and the error values and standard deviations of the EOF pressure, shrinkage along the *X*-axis, and shrinkage along the *Y*-axis are all reduced. After the data are processed with random shuffle, the error values and standard deviation of all outputs are reduced.

## 5. Results and Discussion

Random shuffle and transfer learning for the training of neural networks have been applied for this study. It has been shown that random shuffle can effectively improve the accuracy of the network and reduce the standard deviation. Through observation of the loss value, the vibration amplitude of the loss value of the verification data decreased significantly after using random shuffle, while transfer learning can transfer the learned knowledge to new applications. By expanding weight and bias, it can be found that the shape of weight and bias after using transfer learning is similar. Compared with the initial weight and bias group, which never used transfer learning, the difference is obvious.

### 5.1. Discussion on Random Shuffle Method’s Effect

The training data has been divided into many data sets after implementing random shuffle. Therefore, the data contained in each set are incomplete and only contain partial learning content. The training process of random shuffle data can be regarded as a kind of pretraining; the fragmented data helps the weight and bias of the network to be unable to escape from the local minimum after initialization. The effect is similar to dropout. Taking the training of the 1024 sets of CAE data as an example, the loss value with and without random shuffle implemented are compared; Figure 16 shows the change in loss over 50 training runs. The loss values of the two are similar, and the loss of the verification data is volatile; Figure 17 shows the change over 100 training runs, and the amplitude of the loss value of the verification data is reduced. Figure 18 shows that change over 200 training runs. The loss value of the training data decreases faster for the sets used within the same training runs, and the loss vibration amplitude of the verification data is reduced to only half of the control group. The loss vibration amplitude of the verification data is small, meaning that the network is more stable for forecasting data.

### 5.2. Discussion on the Effect of Transfer Learning

In this section, the learning results using random shuffle for transfer learning are discussed. From Table 24, it can be seen that the error values trained directly from the square plate CAE data and the round plate trial data are not ideal; even with the data processing method of random shuffle, which can optimize standard deviation, the error value is still not ideal. Without changing any hyperparameters, the trained parameters are imported into the network before training. These results are shown in Table 25. The error values for the square and round plate trials have improved significantly. The original warpage along *Z*-axis of the square plate was as high as 56.25, because warpage is complicated, and the plastic flow method adopted, holding pressure, holding time, and cooling time will all affect warpage. Even experts cannot judge such trends in warpage, certainly from the 27 sets of training data, not to mention having an accurate prediction, and after importing the 243 sets for round flat data training weight and bias, the network itself already contains warpage-related trends, and then it has been fine-tuned for the square flat training data, resulting in the amount of warpage along *Z*-axis with a reduction of the error value of the standard deviation to 31.05%.

In the actual reality trials for circular plate, after importing the optimized weight and bias from 1024 sets of circular plate data, the error value of the EOF pressure, shrinkage along the *X*-axis, and shrinkage along the *Y*-axis is reduced to less than 5%. Figure 19 shows the structure of an optimized weight and bias; Table 26, Table 27 and Table 28 show optimized weight and bias. Taking this result as an example, the weight and bias matrix between the two layers are expanded into a column and a distribution map is drawn, as shown in Figure 20, Figure 21, Figure 22, Figure 23, Figure 24 and Figure 25; therein, CAE represents the original distribution before the transfer, TL represents the distribution after transfer, and EXP represents the distribution after direct training. It can be found that the weight and bias distributions of CAE and TL are almost the same, with only a numerical difference. EXP can be different in trend from the other two. Because there are only 16 sets of data for training, the distribution of weight and bias is only between [2, −2], and the other two sets are widely distributed between [6, −6].

It can be understood from the expanded distribution graph that the distribution trend of weight and bias after the transfer has been fixed, and subsequent retraining will only change the distribution by a small margin without major changes, so there is no need for excessive training data and training runs for a usable network to be obtained quickly. Moreover, even a high number of neurons will not help a set of training data that is too small.

## 6. Conclusions

In this study, an artificial neural network has been utilized to render result predictions for the injection molding process. The CAE analysis data are used as training data and the error value is reduced to less than 5% through the Taguchi method and random shuffle method, which we have introduced herein. The performance of the network transfer to the actual machine can show that transfer learning can solve the problem of difficult data acquisition, just like Jiahuan [18] and Yannik [19]. In addition, predicting the performance of the task from the circle plate can support the argument that the transfer learning proposed by Shin [20] can maintain good performance and that its performance is better than the initial training network. In this conclusion, the applications of the neural network and transfer learning are summarized.

### 6.1. Artificial Neural Network Applications

In this study, artificial neural network training was carried out a total of four times, including via use of 243 sets of circular flat CAE data, 27 sets of square flat CAE data, 1024 sets of circular flat CAE data, and 16 sets of circular flat real data. Regarding optimization of the hyperparameters for the network, these are summarized in the following points:Random shuffle can reduce the error rate and standard deviation. In this study, the data volume was 80% and the learning rate was 0.1 for random shuffle pretraining.The Taguchi method can effectively optimize the hyperparameters of artificial neural networks, and after ANOVA analysis, optimal solutions can be achieved.By monitoring the loss value of the training and verification data simultaneously, it can be judged whether there is overfitting. If the loss curve of the training data continues to decline while diverging ever further from the loss curve of the verification data, it may be that the training time is too long and overfitting is caused. If the two loss curves can no longer be optimized, there are two possibilities: one is that the learning rate is not suitable, which makes it impossible to get rid of the local minimum. The other is that the content of the training data is more complex, and the number of neurons is insufficient. Optimization has bottlenecks.

### 6.2. Transfer Learning Applications

This study carried out transfer learning twice, namely the transfer of CAE round plate to CAE square plate and the transfer of CAE round plate to the actual reality trial for round plate. Although both are transfer learning, the former is not using the same model. The time and space for the latter when it comes to data collection are different. Based on the results of these two transfer learning results, this study indicates the following points, summarized here:Since the cost of injection data acquisition is quite high, the transfer learning can predict with less data under some specific conditions. At the same time, actual experiments can prove that transfer learning has better effects in similar work.From the weight and bias distributions before and after transfer learning, it can be found that retraining will not significantly change the distribution; however, a slight change is possible. Therefore, the original network selected will determine the results of transfer learning.If the set of training data is too small to contain enough effective content, the fluctuation range of weight and bias will be smaller. In this state, adding neurons will not improve the training error value.

## Figures and Tables

**Figure 1 polymers-13-03874-f001:**
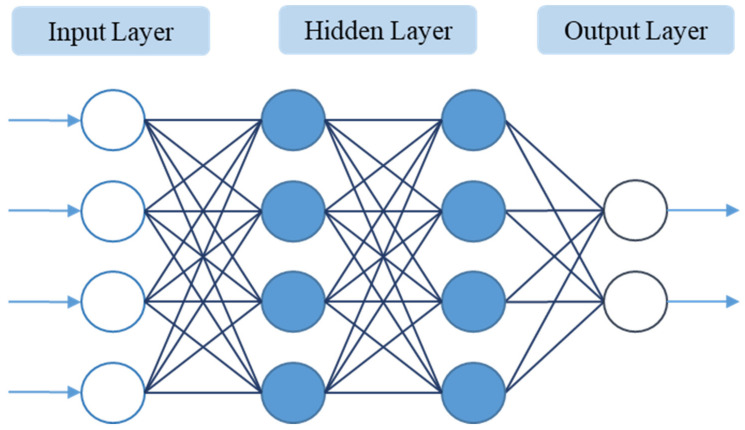
Artificial neural network architecture.

**Figure 2 polymers-13-03874-f002:**
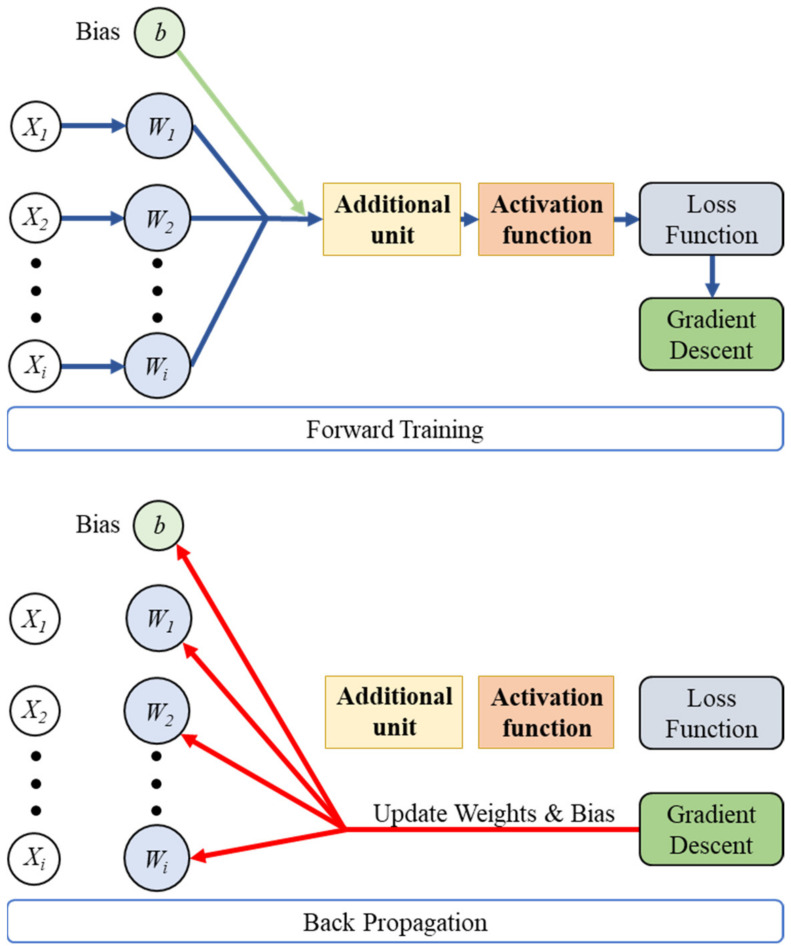
Basic architecture of a backpropagated neural network (BPNN).

**Figure 3 polymers-13-03874-f003:**
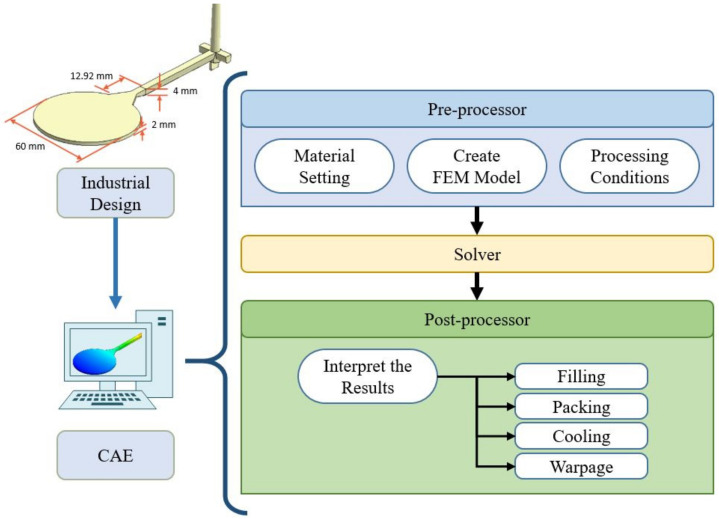
Flowchart of CAE design analysis.

**Figure 4 polymers-13-03874-f004:**
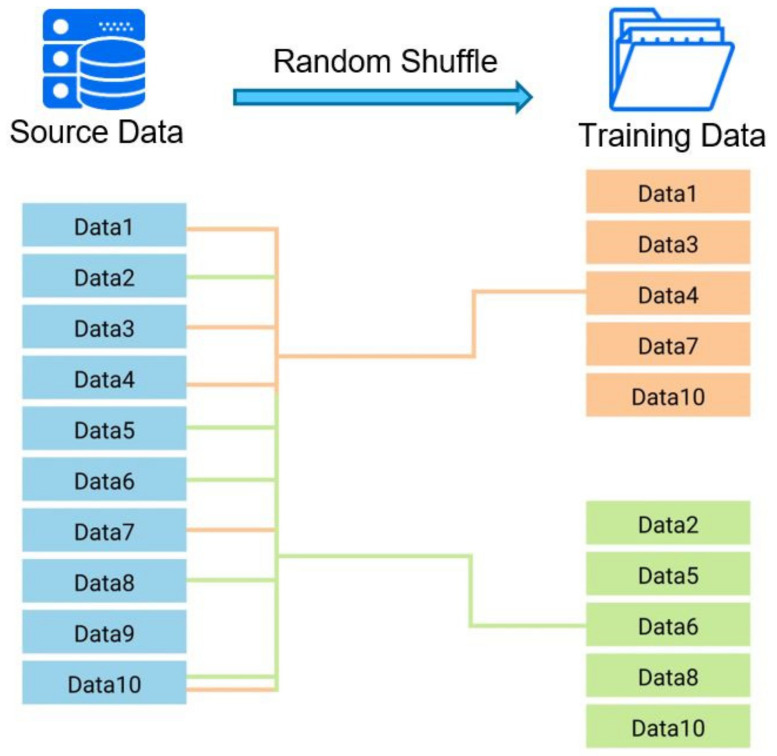
Schematic diagram of random shuffle.

**Figure 5 polymers-13-03874-f005:**
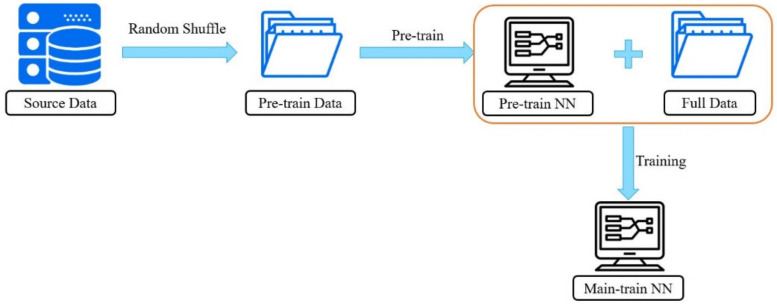
The random shuffle training process.

**Figure 6 polymers-13-03874-f006:**
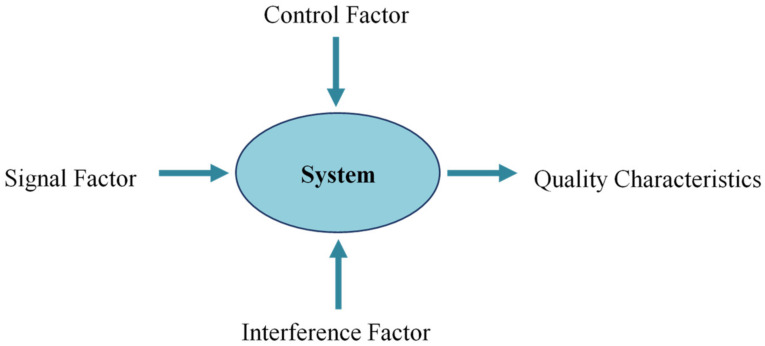
Experimental factors in Taguchi designs.

**Figure 7 polymers-13-03874-f007:**
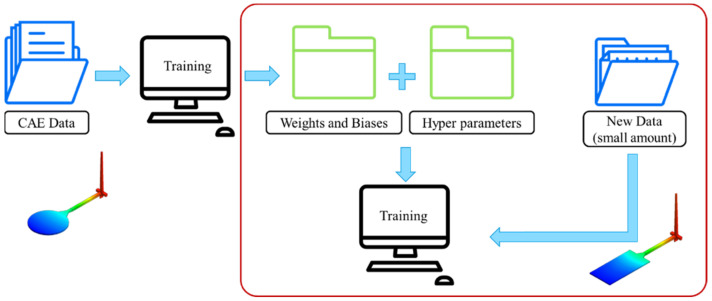
Transfer learning process between different models.

**Figure 8 polymers-13-03874-f008:**
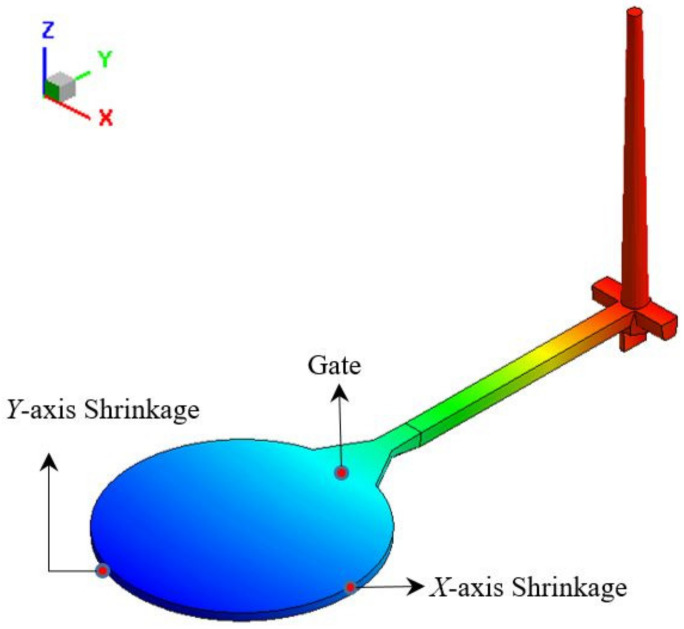
CAE data measurement location of the circular flat model.

**Figure 9 polymers-13-03874-f009:**
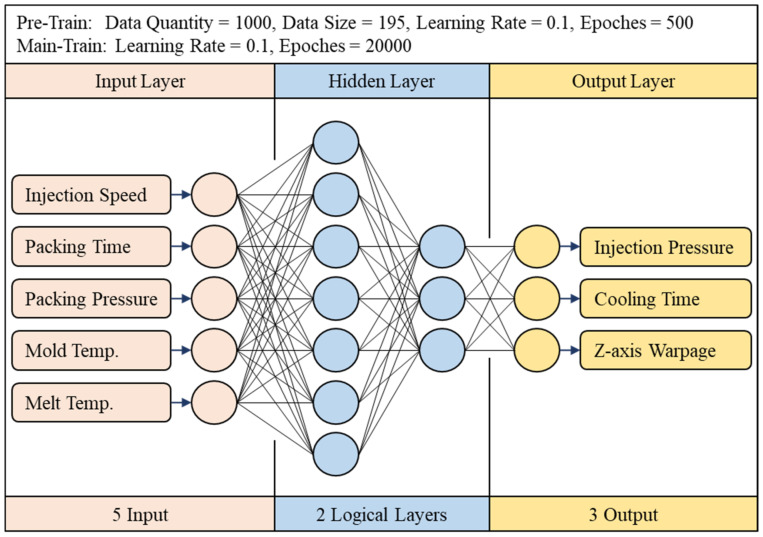
Optimal network architecture (CAE-243).

**Figure 10 polymers-13-03874-f010:**
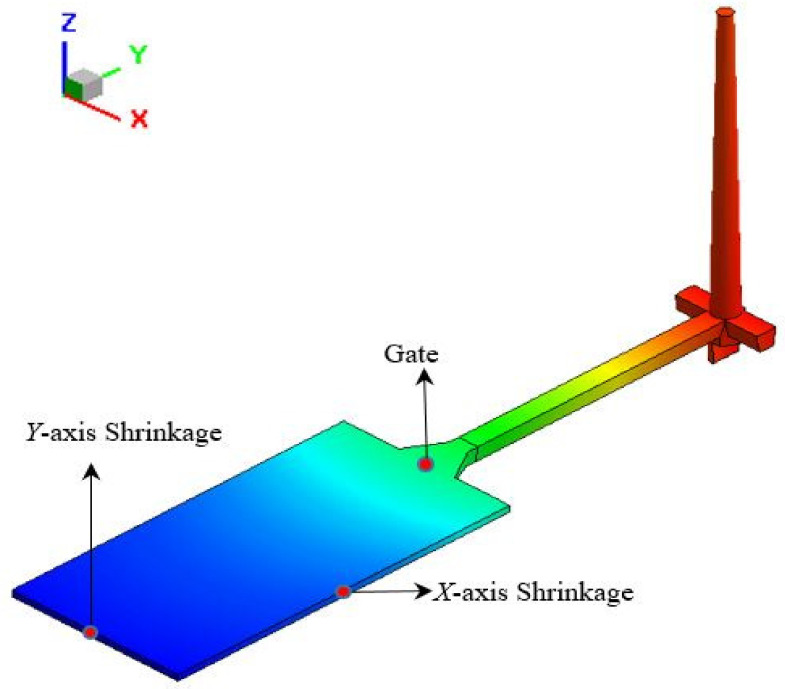
Model measurement position for the square plate.

**Figure 11 polymers-13-03874-f011:**
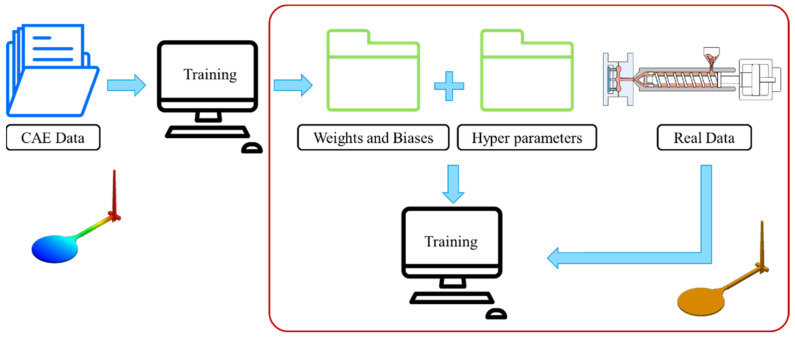
Flow chart of transfer learning from virtual to actual reality trials.

**Figure 12 polymers-13-03874-f012:**
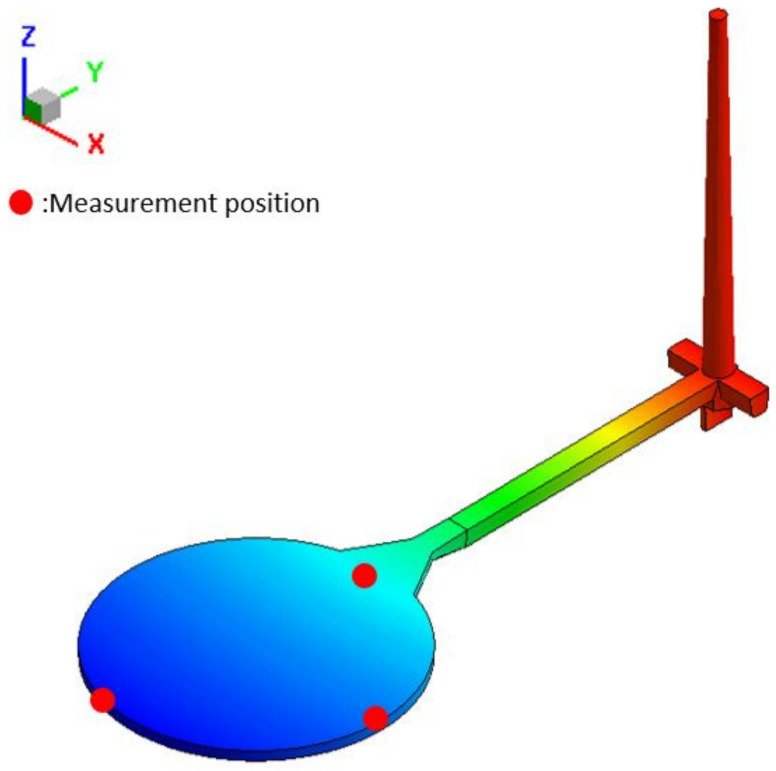
Measurement position of the round plate model.

**Figure 13 polymers-13-03874-f013:**
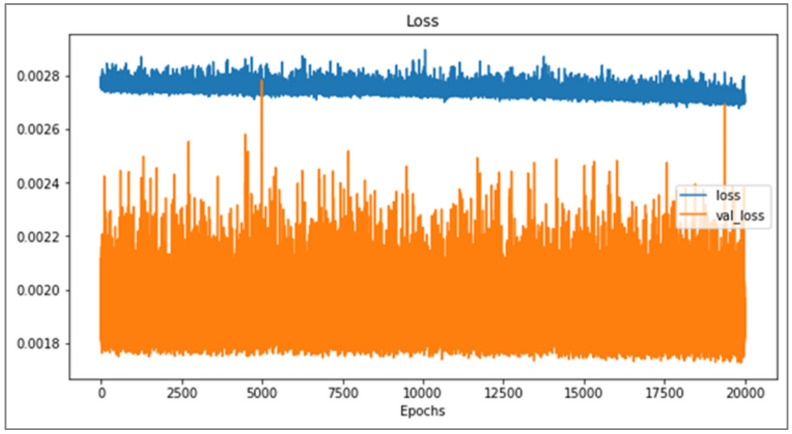
Comparison diagram for loss value in the training set and the verification set.

**Figure 14 polymers-13-03874-f014:**
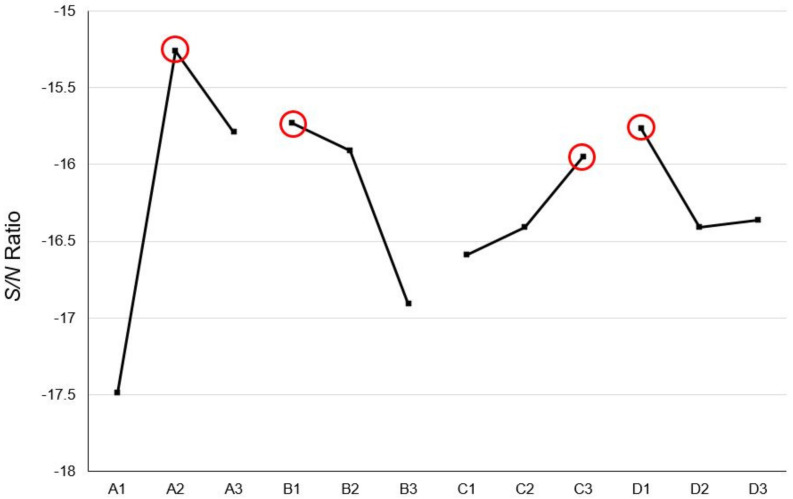
Response graph of *S/N* ratio (CAE-1024).

**Figure 15 polymers-13-03874-f015:**
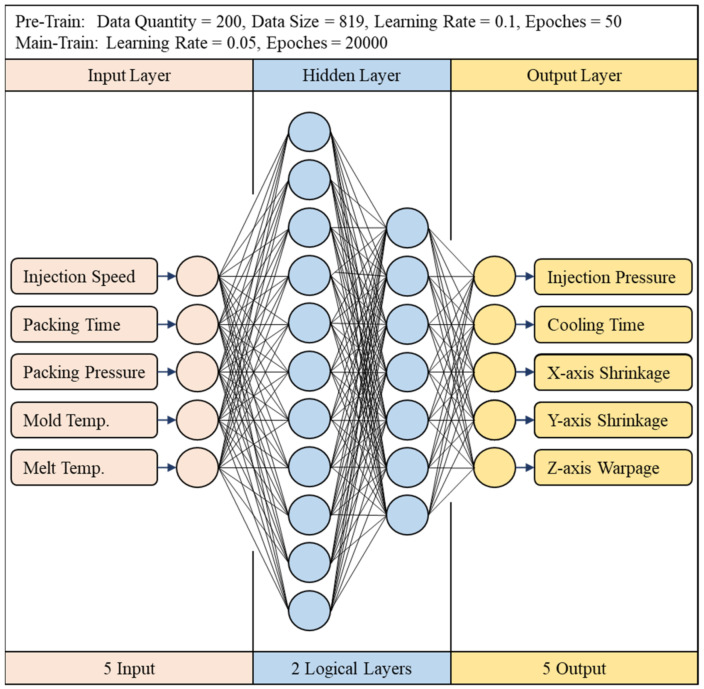
Optimal network architecture (CAE-1024).

**Figure 16 polymers-13-03874-f016:**
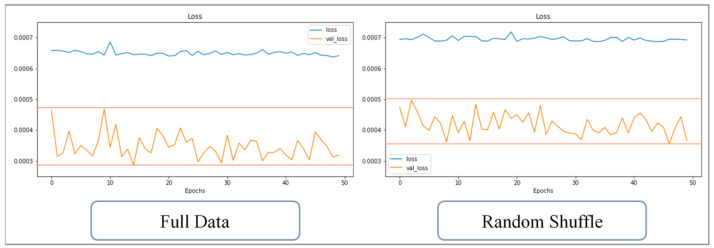
Change of loss value with or without random shuffle (50 runs).

**Figure 17 polymers-13-03874-f017:**
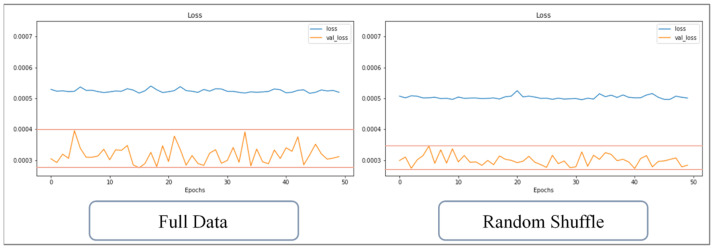
Change of loss value with or without random shuffle (100 runs).

**Figure 18 polymers-13-03874-f018:**
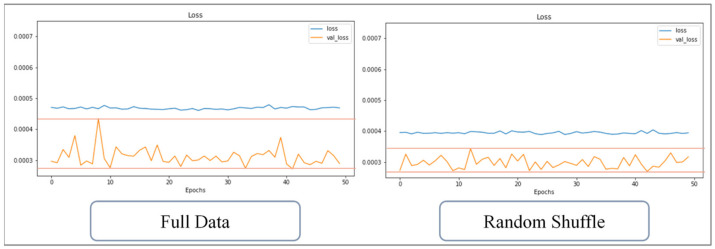
Change of loss value with or without random shuffle (200 runs).

**Figure 19 polymers-13-03874-f019:**
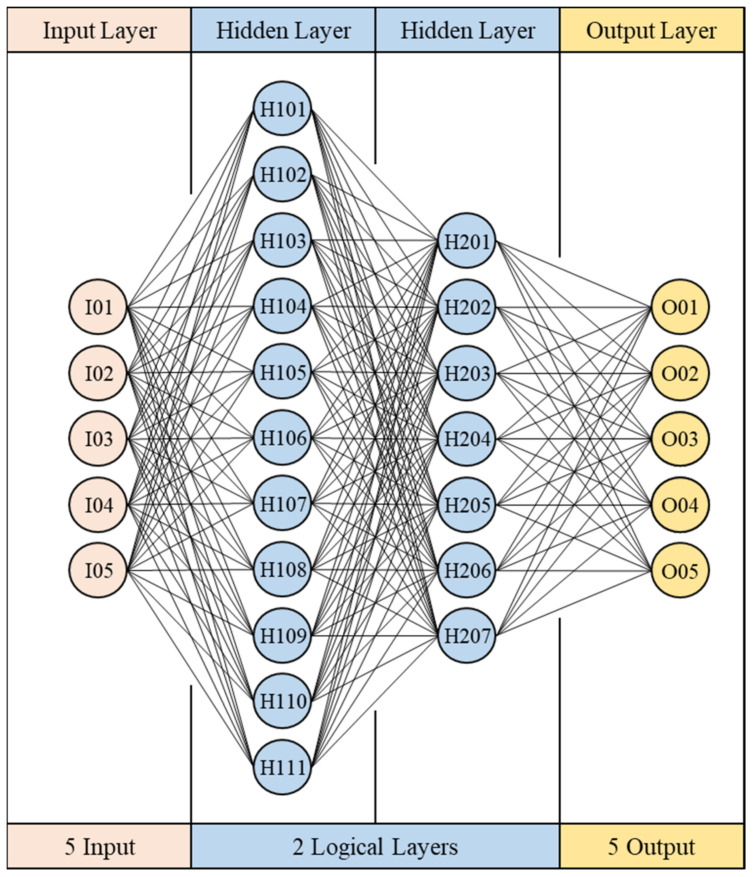
Optimized hyperparameters and network structure for the circular plate (CAE-1024).

**Figure 20 polymers-13-03874-f020:**
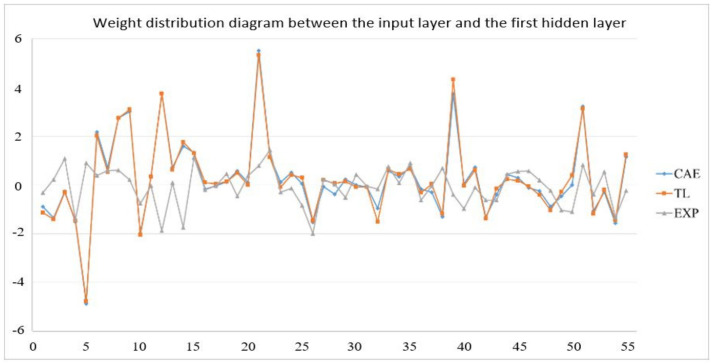
Weight distribution diagram between the input layer and the first hidden layer.

**Figure 21 polymers-13-03874-f021:**
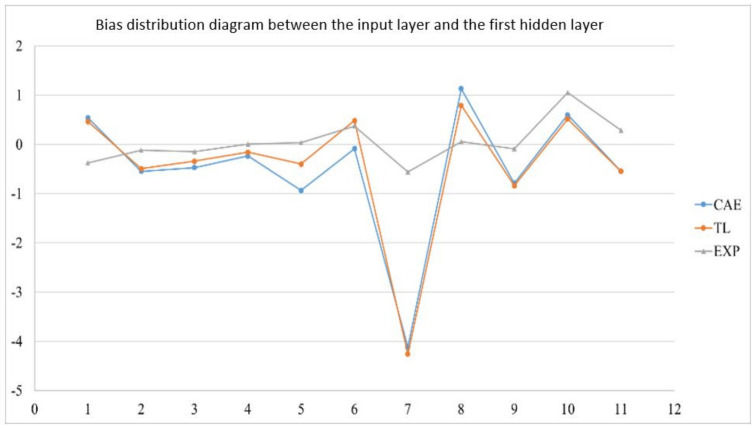
Bias distribution diagram between the input layer and the first hidden layer.

**Figure 22 polymers-13-03874-f022:**
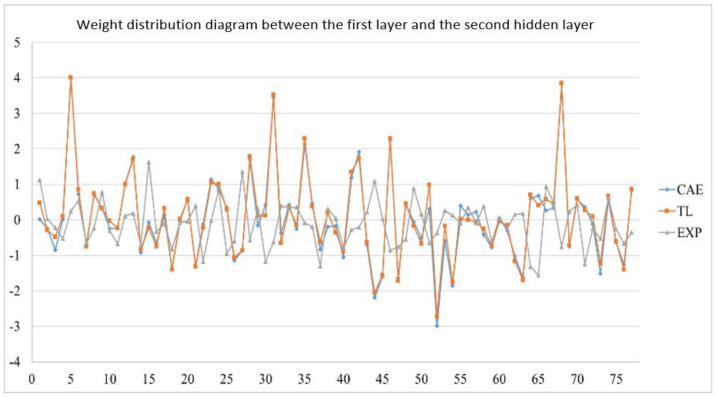
Weight distribution diagram between the first layer and the second hidden layer.

**Figure 23 polymers-13-03874-f023:**
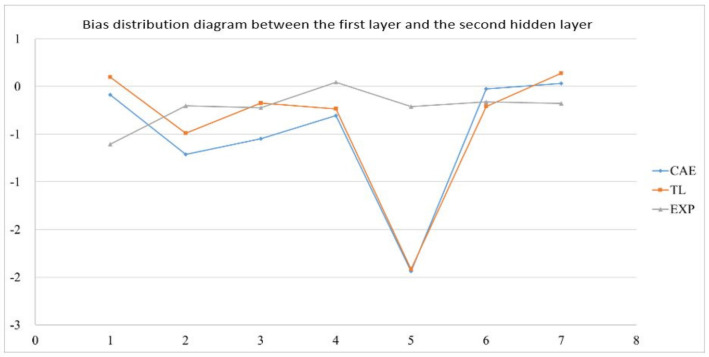
Bias distribution diagram between the first layer and the second hidden layer.

**Figure 24 polymers-13-03874-f024:**
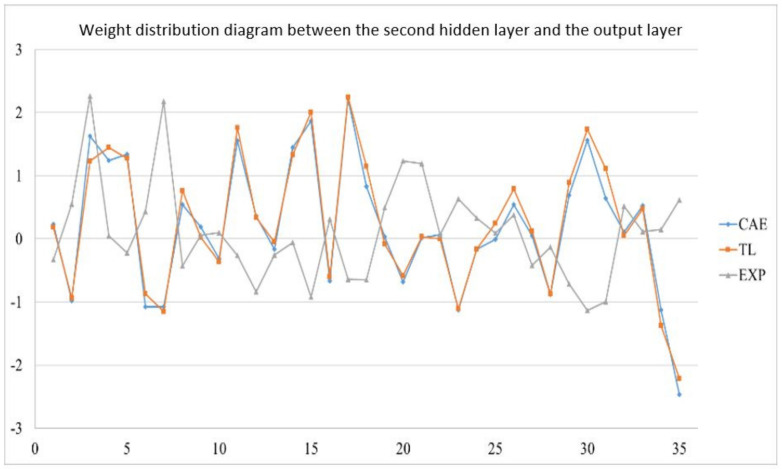
Weight distribution diagram between the second hidden layer and the output layer.

**Figure 25 polymers-13-03874-f025:**
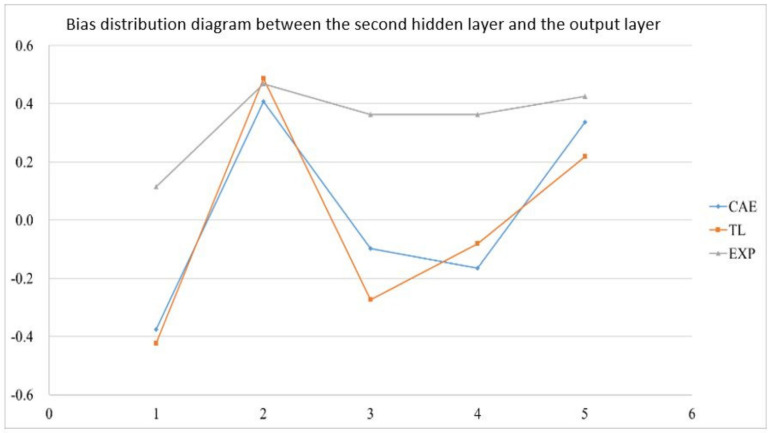
Bias distribution diagram between the second hidden layer and the output layer.

**Table 1 polymers-13-03874-t001:** Training data factors and level settings (CAE-243).

Factor	Default	Level		
Melt Temperature (°C)	210	185	207.5	230
Packing Time (sec)	4.5	3	6	9
Packing Pressure (MPa)	135	100	130	160
Injection Speed (mm/sec)	70	50	65	80
Mold Temperature (°C)	50	40	55	70
Total Processed Data	243			

**Table 2 polymers-13-03874-t002:** Validation data settings (CAE-243).

No.	Melt Temp. (°C)	Packing Time (sec)	Packing Pressure (MPa)	Injection Speed (mm/sec)	Mold Temp. (°C)
1	195	4	110	55	45
2	195	5	120	60	50
3	195	7	140	70	60
4	195	8	150	75	65
5	205	4	120	70	65
6	205	5	110	75	60
7	205	7	150	55	50
8	205	8	140	60	45
9	210	4	140	75	50
10	210	5	150	70	45
11	210	7	110	60	65
12	210	8	120	55	60
13	220	4	150	60	60
14	220	5	140	55	65
15	220	7	120	75	45
16	220	8	110	70	50

**Table 3 polymers-13-03874-t003:** Network hyperparameter settings (CAE-243).

Parameters	Value	Parameters	Value
Epoch	20,000	Learning Rate	0.1
Hidden Layer 1	7	Initial Weight	Random
Hidden Layer 2	3	Initial Bias	Random
Optimized Method	SGD	Activation Function	Sigmoid

**Table 4 polymers-13-03874-t004:** Square tablet training data (CAE-27).

No.	Melt Temp(°C)	Packing Time (sec)	Packing Pressure (MPa)	Injection Speed (mm/sec)	Mold Temp. (°C)
1	185	3	100	50	40
2	185	3	100	50	55
3	185	3	100	50	70
4	185	6	130	65	40
5	185	6	130	65	55
6	185	6	130	65	70
7	185	9	160	80	40
8	185	9	160	80	55
9	185	9	160	80	70
10	207.5	3	130	80	40
11	207.5	3	130	80	55
12	207.5	3	130	80	70
13	207.5	6	160	50	40
14	207.5	6	160	50	55
15	207.5	6	160	50	70
16	207.5	9	100	65	40
17	207.5	9	100	65	55
18	207.5	9	100	65	70
19	230	3	160	65	40
20	230	3	160	65	55
21	230	3	160	65	70
22	230	6	100	80	40
23	230	6	100	80	55
24	230	6	100	80	70
25	230	9	130	50	40
26	230	9	130	50	55
27	230	9	130	50	70

**Table 5 polymers-13-03874-t005:** CAE verification data for square tablet (CAE-27).

No.	Melt Temp. (°C)	Packing Time (sec)	Packing Pressure (MPa)	Injection Speed (mm/sec)	Mold Temp. (°C)
1	195	4	110	55	45
2	195	5	120	60	50
3	195	7	140	70	60
4	195	8	150	75	65
5	205	4	120	70	65
6	205	5	110	75	60
7	205	7	150	55	50
8	205	8	140	60	45
9	210	4	140	75	50
10	210	5	150	70	45
11	210	7	110	60	65
12	210	8	120	55	60
13	220	4	150	60	60
14	220	5	140	55	65
15	220	7	120	75	45
16	220	8	110	70	50

**Table 6 polymers-13-03874-t006:** Hyperparameters settings for the square plate (CAE-27).

Parameters	Value	Parameters	Value
Epoch	20,000	Learning Rate	0.1
Hidden Layer 1	7	Initial Weight	Transfer
Hidden Layer 2	3	Initial Bias	Transfer
Optimized Method	SGD	Activation Function	Sigmoid

**Table 7 polymers-13-03874-t007:** CAE data training results for the round flat plate model (CAE-243).

Circle Plate Result (CAE-243)						
		EOF Pressure	Cooling Time	*Z*-Axis Warpage	*X*-Axis Shrinkage	*Y*-Axis Shrinkage
Full Data	AVG (%)	12.67	8.78	29.84	14.80	13.36
	STD	5.29	4.48	45.46	18.33	11.71
Random Shuffle	AVG (%)	11.22	8.61	19.89	12.83	10.72
	STD	5.22	4.42	26.45	13.91	7.54
Difference	AVG (%)	1.45	0.16	9.94	2.06	2.63
	STD	0.07	0.05	19.01	4.42	4.17

**Table 8 polymers-13-03874-t008:** Training results for the square flat model (CAE-27).

Square Plate Result (CAE-27)						
		EOF Pressure	Cooling Time	*Z*-Axis Warpage	*X*-Axis Shrinkage	*Y*-Axis Shrinkage
Full Data	AVG (%)	4.85	10.98	59.61	17.14	20.64
	STD	2.18	7.43	66.65	22.05	18.43
Random Shuffle	AVG (%)	3.91	10.16	56.25	15.39	18.54
	STD	1.90	6.40	60.58	18.66	17.63
Difference	AVG (%)	0.94	0.82	3.36	1.75	2.10
	STD	0.28	1.03	6.07	3.39	0.81

**Table 9 polymers-13-03874-t009:** Transfer learning results for the square flat model (CAE-27).

Square Plate with Transfer Learning Result (CAE-27)						
		EOF Pressure	Cooling Time	*Z*-Axis Warpage	*X*-Axis Shrinkage	*Y*-Axis Shrinkage
Full Data	AVG (%)	3.44	8.69	79.96	15.02	18.15
	STD	1.83	5.36	70.34	10.85	12.18
Random Shuffle	AVG (%)	2.77	8.48	31.05	11.81	16.46
	STD	1.80	4.94	17.56	8.97	11.19
Difference	AVG (%)	0.67	0.22	48.91	3.22	1.69
	STD	0.03	0.42	52.78	1.89	0.99

**Table 10 polymers-13-03874-t010:** Training data for the round plate (CAE-1024).

Factor	Default	Level			
Melt Temperature (°C)	210	185	200	215	230
Packing Time (sec)	4.5	3	5	7	9
Packing Pressure (MPa)	135	100	120	140	160
Injection Speed (mm/sec)	70	50	60	70	80
Mold Temperature (°C)	50	40	50	60	70
Total Process Data	1024				

**Table 11 polymers-13-03874-t011:** Validation data for the round plate (CAE-1024).

No.	Melt Temp. (°C)	Packing Time (sec)	Packing Pressure (MPa)	Injection Speed (mm/sec)	Mold Temp. (°C)
1	194	4.2	112	56	46
2	194	5.4	124	62	52
3	194	6.6	136	68	58
4	194	7.8	148	74	64
5	203	4.2	124	68	64
6	203	5.4	112	74	58
7	203	6.6	148	56	52
8	203	7.8	136	62	46
9	212	4.2	136	74	52
10	212	5.4	148	68	46
11	212	6.6	112	62	64
12	212	7.8	124	56	58
13	221	4.2	148	64	58
14	221	5.4	136	56	64
15	221	6.6	124	74	46
16	221	7.8	112	68	52

**Table 12 polymers-13-03874-t012:** Artificial neural network factors and level settings.

Factor	Parameters	Level 1	Level 2	Level 3
A	Training Cycle	10,000	20,000	30,000
B	Learning Rate	0.05	0.1	0.3
C	Hidden Layer 1	7	9	11
D	Hidden Layer 2	7	9	11

**Table 13 polymers-13-03874-t013:** L9 orthogonal table (CAE-1024).

No	A	B	C	D	Training Cycle	Learning Rate	Hidden Layer 1	Hidden Layer 2
1	1	1	1	1	10,000	0.05	7	7
2	1	2	2	2	10,000	0.1	9	9
3	1	3	3	3	10,000	0.3	11	11
4	2	1	2	3	20,000	0.05	9	11
5	2	2	3	1	20,000	0.1	11	7
6	2	3	1	2	20,000	0.3	7	9
7	3	1	3	2	30,000	0.05	11	9
8	3	2	1	3	30,000	0.1	7	11
9	3	3	2	1	30,000	0.3	9	7

**Table 14 polymers-13-03874-t014:** Experimental results from the Taguchi orthogonal array.

No	A	B	C	D	EOF Pressure	Cooling Time	*Z*−Axis Warpage	*X*−Axis Radius	*Y*−Axis Radius	*S/N* Ratio
1	1	1	1	1	5.04	3.21	7.10	2.22	1.99	−17.03
2	1	2	2	2	4.90	3.20	7.31	2.28	2.18	−17.28
3	1	3	3	3	4.97	3.55	8.10	2.28	2.22	−18.17
4	2	1	2	3	4.98	1.36	5.50	2.17	2.56	−14.81
5	2	2	3	1	5.10	1.72	5.22	2.04	2.20	−14.35
6	2	3	1	2	4.78	3.42	6.78	2.11	2.38	−16.62
7	3	1	3	2	5.04	1.32	5.85	1.98	1.96	−15.34
8	3	2	1	3	5.18	2.06	6.39	1.73	2.40	−16.11
9	3	3	2	1	4.94	1.13	6.26	2.08	2.09	−15.93

**Table 15 polymers-13-03874-t015:** ANOVA analysis (CAE-1024).

	A	B	C	D
LEVEL 1	−17.49	−15.73	−16.59	−15.77
LEVEL 2	−15.26	−15.91	−16.41	−16.41
LEVEL 3	−15.79	−16.91	−15.95	−16.36

**Table 16 polymers-13-03874-t016:** Verification of optimized hyperparameters (CAE-1024).

	Training Cycle	Learning Ratio	Layer 1	Layer 2	*Z*-axis
No.5	20,000	0.1	11	7	5.22%
Op1.	20,000	0.05	11	7	3.61%
Op2.	20,000	0.03	13	5	4.88%

**Table 17 polymers-13-03874-t017:** Optimized network hyperparameters (CAE-1024).

Parameters	Value	Parameters	Value
Epoch	20,000	Learning Rate	0.05
Hidden Layer 1	11	Initial Weight	Random
Hidden Layer 2	7	Initial Bias	Random
Optimized Method	SGD	Activation Function	Sigmoid

**Table 18 polymers-13-03874-t018:** Training data for the round plate model (EXP-16).

No.	Melt Temp. (°C)	Packing Time (sec)	Packing Pressure (MPa)	Injection Speed (mm/sec)	Mold Temp. (°C)
1	185	3	100	50	40
2	185	5	113	60	50
3	185	7	126	70	60
4	185	9	139	80	70
5	200	3	113	70	70
6	200	5	100	80	60
7	200	7	139	50	50
8	200	9	126	60	40
9	215	3	126	80	50
10	215	5	139	70	40
11	215	7	100	60	70
12	215	9	113	50	60
13	230	3	139	60	60
14	230	5	126	50	70
15	230	7	113	80	40
16	230	9	100	70	50

**Table 19 polymers-13-03874-t019:** Validation data for the round plate model (EXP-16).

No.	Melt Temp. (°C)	Packing Time (sec)	Packing Pressure (MPa)	Injection Speed (mm/sec)	Mold Temp. (°C)
1	194	4.2	109	56	46
2	194	5.4	118	62	52
3	194	6.6	127	68	58
4	194	7.8	136	74	64
5	203	4.2	118	68	64
6	203	5.4	109	74	58
7	203	6.6	136	56	52
8	203	7.8	127	62	46
9	212	4.2	127	74	52
10	212	5.4	136	68	46
11	212	6.6	109	62	64
12	212	7.8	118	56	58
13	221	4.2	136	62	58
14	221	5.4	127	56	64
15	221	6.6	118	74	46
16	221	7.8	109	68	52

**Table 20 polymers-13-03874-t020:** Network hyperparameter settings (EXP-16).

Parameters	Value	Parameters	Value
Epoch	20,000	Learning Rate	0.05
Hidden Layer 1	11	Initial Weight	Transfer
Hidden Layer 2	7	Initial Bias	Transfer
Optimized Method	SGD	Activation Function	Sigmoid

**Table 21 polymers-13-03874-t021:** Training results for the round flat model (CAE-1024).

Circle Plate Result (CAE-1024)						
		EOF Pressure	Cooling Time	*Z*-Axis Warpage	*X*-Axis Shrinkage	*Y*-Axis Shrinkage
Full Data	AVG (%)	4.01	1.84	8.50	3.48	2.73
	STD	2.61	1.35	5.21	2.78	2.23
Random Shuffle	AVG (%)	3.99	1.27	3.61	1.34	1.89
	STD	2.01	0.92	2.87	1.01	1.10
Difference	AVG (%)	0.02	0.57	4.9	2.14	0.84
	STD	0.60	0.43	2.34	1.77	1.13

**Table 22 polymers-13-03874-t022:** Training results for the round flat model (EXP-16).

Circle Plate Result (EXP-16)						
		EOF Pressure	Cooling Time	*Z*-Axis Warpage	*X*-Axis Shrinkage	*Y*-Axis Shrinkage
Full Data	AVG (%)	15.03	8.31	9.26	15.03	8.31
	STD	8.58	13.34	15.73	8.58	13.34
Random Shuffle	AVG (%)	13.61	7.36	7.45	13.61	7.36
	STD	6.62	9.33	8.63	6.62	9.33
Difference	AVG (%)	1.42	0.95	1.81	1.42	0.95
	STD	4.96	4.01	7.10	4.96	4.01

**Table 23 polymers-13-03874-t023:** Results of transfer learning for the round plate model (EXP-16).

Circle Plate Result (EXP-16)		with Transfer Learning			without Transfer Learning		
		EOF Pressure	*X*-Axis Shrinkage	*Y*-Axis Shrinkage	EOF Pressure	*X*-Axis Shrinkage	*Y*-Axis Shrinkage
Full Data	AVG (%)	5.88	2.56	3.96	15.03	8.31	9.26
	STD	4.71	2.26	3.97	8.58	13.34	15.73
Random Shuffle	AVG (%)	5.56	2.35	3.91	13.61	7.36	7.45
	STD	4.20	2.19	3.42	6.62	9.33	8.63
Difference	AVG (%)	0.32	0.21	0.05	1.42	0.95	1.81
	STD	0.51	0.07	0.55	4.96	4.01	7.10

**Table 24 polymers-13-03874-t024:** Predicted results for square plate (CAE-27) and round plate (EXP-16).

Result of Square Plate (CAE-27) & Circle Plate (EXP-16)						
		EOF Pressure	Cooling Time	*Z*-Axis Warpage	*X*-Axis Shrinkage	*Y*-Axis Shrinkage
Square Plate	AVG (%)	3.91	10.16	56.25	15.39	18.54
	STD	1.90	6.40	60.58	18.66	17.63
Circle Plate	AVG (%)	13.61			7.36	7.45
	STD	6.62			9.33	8.63

**Table 25 polymers-13-03874-t025:** Transfer learning results for square plate (CAE-27) and round plate (EXP-16).

Result of Square Plate (CAE-27) & Circle Plate (EXP-16)						
		EOF Pressure	Cooling Time	*Z*-Axis Warpage	*X*-Axis Shrinkage	*Y*-Axis Shrinkage
Square Plate	AVG (%)	3.91	10.16	56.25	15.39	18.54
	STD	1.90	6.40	60.58	18.66	17.63
Square Plate (TL)	AVG (%)	2.77	8.48	31.05	11.81	16.46
	STD	1.80	4.94	17.56	8.97	11.19
Circle Plate	AVG (%)	13.61			7.36	7.45
	STD	6.62			9.33	8.63
Circle Plate(TL)	AVG (%)	5.56			2.35	3.91
	STD	4.20			2.19	3.42

**Table 26 polymers-13-03874-t026:** Weight and bias between the input layer and the first hidden layer (CAE-1024).

Input to Hidden Layer 1											
Weight	H101	H102	H103	H104	H105	H106	H107	H108	H109	H110	H111
I01	−1.1380	−1.3887	−0.2855	−1.4546	−4.7642	2.0215	0.5312	2.7493	3.0976	−2.0409	0.3560
I02	3.7457	0.6322	1.7630	1.2978	0.0940	0.0474	0.1391	0.4969	−0.0035	5.3307	1.1453
I03	−0.0767	0.4169	0.2950	−1.4659	0.2065	0.0649	0.1480	−0.0729	−0.0716	−1.5080	0.6472
I04	0.4478	0.6541	−0.3143	0.0424	−1.1876	4.3344	−0.0358	0.6161	−1.3805	−0.1439	0.2278
I05	0.1851	−0.0521	−0.4043	−1.0403	−0.2708	0.3893	3.1411	−1.1837	−0.1945	−1.4266	1.2472
Bias	0.4616	−0.4964	−0.3441	−0.1590	−0.4003	0.4831	−4.2604	0.7888	−0.8406	0.5141	−0.5477

**Table 27 polymers-13-03874-t027:** Weight and bias between the first and second hidden layers (CAE-1024).

Input to Hidden Layer 2							
Weight	H201	H202	H203	H204	H205	H206	H207
H101	0.4846	−0.2906	−0.4810	0.0960	4.0143	0.8524	−0.7530
H102	0.7461	0.3182	−0.0355	−0.2350	0.9976	1.7191	−0.8385
H103	−0.2167	−0.7549	0.3227	−1.3992	0.0234	0.5743	−1.3188
H104	−0.2193	1.0535	1.0047	0.3022	−1.0671	−0.8595	1.7802
H105	0.1245	0.1110	3.5158	−0.6555	0.3523	−0.1515	2.2811
H106	0.3782	−0.6278	0.1922	−0.3553	−0.9026	1.3512	1.7357
H107	−0.6336	−2.0454	−1.5661	2.2850	−1.7217	0.4527	−0.1687
H108	−0.6807	0.9787	−2.7245	−0.1729	−1.7629	0.0099	−0.0046
H109	−0.0665	−0.2635	−0.7414	−0.0496	−0.1540	−1.1532	−1.7003
H110	0.7065	0.4036	0.5818	0.4633	3.8549	−0.7217	0.6002
H111	0.2691	0.0886	−1.2342	0.6715	−0.6184	−1.4114	0.8704
Bias	0.0969	−0.4892	−0.1725	−0.2370	−1.9194	−0.2119	0.1383

**Table 28 polymers-13-03874-t028:** Weight and bias between the second hidden layer and the output layer (CAE-1024).

Hidden Layer 2 to Output					
Weight	O01	O02	O03	O04	O05
H201	0.1771	−0.9374	1.2355	1.4501	1.2729
H202	−0.8741	−1.1480	0.7641	0.0267	−0.3646
H203	1.7557	0.3356	−0.0501	1.3303	2.0018
H204	−0.6002	2.2428	1.1498	−0.0814	−0.5862
H205	0.0348	0.0010	−1.1131	−0.1650	0.2440
H206	0.7898	0.1205	−0.8733	0.8862	1.7326
H207	1.1126	0.0468	0.4803	−1.3758	−2.2127
Bias	−0.4228	0.4861	−0.2742	−0.0809	0.2185

## Data Availability

Not applicable.

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
