# Peer review of "Transfer Learning Applied to Characteristic Prediction of Injection Molded Products"

_polymers, 2021, doi:10.3390/polym13223874_

Round 1
Reviewer 1 Report
The current work has merit, quite technical but sufficiently novel to highlight the machine learning potential for injection moulding.
The introduction is very dedicated. Perhaps good to start the discussion with examples of injection moulding processes and applications. There are several Polymers contributions to cite. I suggest to add at least three, also including a composite one (e.g. Polymers 2019, 11, 87)
Figure 5 can be improved for a general reader
2.6.1 please make a difference between n and N
Please explain to a general reader the in silico nature of the “experiments”
General comment: the number of reference is quite low. Please make this stronger, specifically in the results and discussion making comparison to previous work.
Author Response
To the reviewers:
Thank you for the suggestions. We have added several references to the paper, and also added the results of transfer learning in injection molding simulation in the conclusion. In addition, we have cited the references suggested by the reviewers. The formula (3) in 2.6.1 is the formula of Taguchi method Quality Characteristics. Last not least, we have redrawn figure 5 to make it easier to read.
Reviewer 2 Report
Attached

Author Response
To the reviewers:
Thank you for the suggestions. We have added several references to verify the effectiveness of transfer learning, and also added the results of transfer learning in injection molding simulation in the conclusion. In addition, we have cited the document suggested by the reviewers. The formula(3) in 2.6.1 is the formula of Taguchi method Quality Characteristics. Last not least, the reasons for using Back Propagation Neural Network (BPNN), please refer to the supplementary explanation in 2.1.1.
Round 2
Reviewer 2 Report
Attached

Author Response
To the reviewer:
Thank you for the suggestions. We have a more detailed description of the Introduction. At the same time, we have redrawn Figure 4 to more clearly express the operating principle of Random Shuffle. In addition, we have cited relevant references in the conclusion to support our research results.